# The Association between Dysbiosis and Neurological Conditions Often Manifesting with Chronic Pain

**DOI:** 10.3390/biomedicines11030748

**Published:** 2023-03-01

**Authors:** Mary Garvey

**Affiliations:** 1Department of Life Science, Atlantic Technological University, F91 YW50 Sligo, Ireland; mary.garvey@atu.ie; Tel.: +353-071-9305529; 2Centre for Precision Engineering, Materials and Manufacturing Research (PEM), Atlantic Technological University, F91 YW50 Sligo, Ireland

**Keywords:** microbiota, intestinal, chronic pain, neuroinflammation, sensitization, intestinal permeability

## Abstract

The prevalence of neurological conditions which manifest with chronic pain is increasing globally, where the World Health Organisation has now classified chronic pain as a risk factor for death by suicide. While many chronic pain conditions have a definitive underlying aetiology, non-somatic conditions represent difficult-to-diagnose and difficult-to-treat public health issues. The interaction of the immune system and nervous system has become an important area in understanding the occurrence of neuroinflammation, nociception, peripheral and central sensitisation seen in chronic pain. More recently, however, the role of the resident microbial species in the human gastrointestinal tract has become evident. Dysbiosis, an alteration in the microbial species present in favour of non-beneficial and pathogenic species has emerged as important in many chronic pain conditions, including functional somatic syndromes, autoimmune disease and neurological diseases. In particular, a decreased abundance of small chain fatty acid, e.g., butyrate-producing bacteria, including *Faecalibacterium*, *Firmicutes* and some *Bacteroides* spp., is frequently evident in morbidities associated with long-term pain. Microbes involved in the production of neurotransmitters serotonin, GABA, glutamate and dopamine, which mediate the gut-brain, axis are also important. This review outlines the dysbiosis present in many disease states manifesting with chronic pain, where an overlap in morbidities is also frequently present in patients.

## 1. Introduction

Chronic pain is pain which persists longer than 3 months past the normal healing time of the tissue [1], as associated with somatic injury. The International Association for the Study of Pain (IASP) now defines chronic pain as “an unpleasant sensory and emotional experience associated with, or resembling that associated with, actual or potential tissue damage” [2]. While pain is associated with many chronic conditions, the most recent update of the International Classification of Diseases (ICD-11), however, now lists chronic pain as its own health condition [3]. There are many categories and types of pain, including neuropathic, nociceptive, musculoskeletal, inflammatory, psychogenic and mechanical [4]. Additional types of chronic pain are categorised into functional somatic syndromes (FSS) where chronic pain persists in the absence of any physical or structural defect in the body or absence of a somatic condition [5]. Chronic pain is multifaceted, having a significant impact on the suffering individual socially, economically and physically, and affects approximately 20% of the global population [2]. Chronic pain is also associated with opioid addiction, mental health conditions, social isolation and reduced quality of life [6]. Importantly, there is a significant rate of moderate and severe mental health conditions associated with chronic or persistent pain [7]. Chronic co-morbidities—either physical and/or mental—are present in 88% of chronic pain patients [1]. Risk factors associated with chronic pain include sociodemographic, clinical, psychological and biological factors (females are more prone than males) [1]. Treatment protocols typically include pain therapy, such as non-steroidal anti-inflammatory drugs, opioids, psychological therapy, physical therapy, or alternative non-pharmacological treatments [8]. Additional therapeutics often prescribed include antiepileptic drugs (gabapentin or pregabalin), antidepressants, e.g., tricyclic antidepressants amitriptyline and serotonin-norepinephrine reuptake inhibitor duloxetine, topical analgesics, e.g., lidocaine, muscle relaxers, N-methyl-d-aspartate receptor antagonists and alpha 2 adrenergic agonists [4]. Nonpharmacological treatment regimens include heat and cold therapy, cognitive behavioural therapy, relaxation therapy, counselling, ultrasound stimulation, acupuncture, aerobic exercise, chiropractic, physical therapy, osteopathic manipulative medicine and occupational therapy [4]. The widespread prescription of opioids raises addiction issues as ca. 20% of chronic pain patients are prescribed opioid pain relief for acute and chronic pain. Opioid analgesics are the mainstay for cancer pain patients but their use in chronic non-cancer pain (CNCP) has limited success [8]. CNCP is moderate or severe pain with long-term duration often associated with osteoarthritis, neuropathic pain, FSSs, e.g., fibromyalgia and autoimmune conditions, e.g., rheumatoid arthritis, amongst other conditions (Table 1) [9]. While the use of opioids for long-term treatment of pain does not appear effective in reducing pain levels [7], the risks associated with opioid use include opioid tolerance, diversion, addiction, overdose and death [10]. Opioid-induced hyperalgesia is associated with increased chronic pain symptoms as it leads to central sensitization, neuroinflammation and worsening pain symptoms in patients prescribed opioid analgesics [11]. Importantly, patients of chronic pain also have increased rates of suicide ideation, attempts and successful suicides [12]. Studies have shown chronic pain sufferers are ca. 2 to 3 times more at risk of suicidal tendencies and suicide completion, the World Health Organisation (WHO) now recognises chronic pain as a risk factor for death by suicide [13]. Suicidal behaviour rates vary between chronic pain patients with 41% of fibromyalgia patients having suicidal behaviours and 17% attempting suicide, fibromyalgia is associated with a 10 times higher suicide rate than non-chronic pain persons [13]. As research aims to identify the causation of chronic pain, particularly in non-somatic cases, there is increasing focus on the role of the gastrointestinal microbiota, microbiota neurological interactions and intestinal dysbiosis on the manifestation of chronic pain. The microbiota housed in the gastrointestinal tract (GIT) is a complex and dynamic community of microbial species predominately bacteria, which have essential and important roles in maintaining homeostasis, health and wellbeing [14]. This review aims to highlight the impact of an imbalance in this community and alterations on microbial diversity on chronic pain conditions. Interactions between the GIT microbiota and the central nervous system (CNS), neuroinflammation, neuroexcitation and neurogenic pain will be discussed in terms of chronic pain conditions.

## 2. Dysbiosis—Imbalance in the Gut

The microbiota is a vast array of commensal or mutualistic microorganisms, including bacteria, bacteriophages, viruses and fungi, which co-exist in and on human body cavities [20]. The majority of species are bacterial and are predominately located in the GIT, where they contribute to homeostasis and the nutritional status of the host (Table 2). The bacterial microbiota is dominated by the phyla *Firmicutes*, *Actinobacteria* and *Bacteroidetes* with lower relative abundances of *Verrucomicrobia* and *Proteobacteria* [21]. Importantly, the gut microbiota, and its gene pool termed microbiome, is key to host immunity, regulating GIT endocrine function, neurological signalling, digesting food, xenobiotic modification, metabolism, elimination and producing beneficial compounds for the human host [22]. For example, certain species of *Bifidobacterium* of the *Actinobacteria* phylum are associated with anti-inflammatory properties and decreased intestinal permeability [23]. Interestingly, microbial metabolites produced by these resident species can also affect the expression of key liver enzymes, including the cytochrome P450 (CYP) superfamily influencing the metabolism of xenobiotics by the liver [21]. The core metabolic functions of the microbiota and microbiome benefiting the host include glycosaminoglycan degradation, fermentation of complex polysaccharides to generate short chain fatty acids (SCFAs) and the biosynthesis of some essential amino acids and vitamins [20]. For example, the GIT microbiota produce many of the B-vitamins, including vitamin B9 an B12 required for normal host function [24], and forms of vitamin K, termed menaquinones, which may have anti-inflammatory activity [25]. Amino-acid-fermenting bacteria in the GIT include bacteria *Clostridia*, the *Bacillus–Lactobacillus–Streptococcus* groups, *Proteobacteria* and *Peptostreptococcus*, which impact amino acid absorption [26]. The microbiota also express carbohydrate-active enzymes, allowing them to ferment complex carbohydrates and fibre generating metabolites, such as SCFAs [14]. SCFAs are known to improve the gut health via maintenance of intestinal barrier integrity, mucus production, protection against inflammation and to reduce the risk of colorectal cancer [27]. Additionally, certain SCFAs can cross the blood–brain barrier (BBB), regulate the BBB permeability, and re-establish damaged microglia by restoring their function and morphology [20]. Importantly, SCFA’s provide approximately 50% of the energy requirements of epithelial cells [28]. Firmicutes forms the largest percentage of human gut microbiota having 200 genera and many species that produce butyrate in the GIT [29]. Intestinal barrier integrity is essential to main permeability, prevent leakage of intestinal content and translocation to the systemic circulation. Intestinal permeability leads to a translocation of gut microbes and their neuroactive metabolites and components, which can promote a neuroinflammatory response in the CNS and peripheral nervous system (PNS) [30]. The composition of the microbiota varies at different locations of the GIT and is affected by oxygen, chemical, endogenous antimicrobials, nutritional and immunological features of the gut [14]. Dysbiosis of this system is a change in the resident microbiota and reduced microbial diversity, which disrupts gut homeostasis, ultimately having varied negative impacts on the host [11]. Alterations in GIT microbiota impacts immune system activity, metabolic disorders, including diabetes, insulin resistance, inflammation and dyslipidaemia, and impacts endocrine regulation [31]. An imbalance in the gut microbiota and microbiome has been established in the aetiology of many intestinal and extra intestinal diseases, including inflammatory bowel diseases (IBD) coeliac disease, irritable bowel syndrome (IBS), colon cancer, liver and pancreas disorders [32]. Dysbiosis results in leaky gut, visceral hypersensitivity, immune activation, inflammation, mood disorders and chronic fatigue in patients [5]. Research suggests that increased abundance of *Parcubacteria* and lower *Verrucomicrobia* results in neuropathic pain and anhedonia (loss of pleasure) in test rats [33]. The species *Akkermansia muciniphila* of the phylum *Verrucomicrobia* may be associated with pain inhibition [34]. *A. muciniphila* also plays a role in controlling immunological function, enhances the synthesis of antimicrobial peptides and improves gut homeostasis [35]. Many of the intestinal microbes produce neurotransmitters, including acetylcholine produced by *Lactobacillus plantarum*, dopamine produced by *Proteus vulgaris*, *Bacillus*, and *Serratia marcescens*, GABA produced by *Lactobacillus* and *Bifidobacterium*, histamine produced by *Citrobacter* and *Enterobacter*, norepinephrine produced by *Saccharomyces*, *Bacillus*, and *E. coli* and serotonin (5-HT) by *E. coli*, *Enterococcus*, *Candida* and *Streptococcus* [36]. Therefore, the reduced microbial diversity observed in dysbiosis alters the expression of neurotransmitters affecting the local enteric nervous system and the CNS [5].

### 2.1. Causation of Dysbiosis

The gut microbiota is unique to each individual and influenced by mode of birth, infant feeding, lifestyle and dietary choices, medication and the genetics of the host. A healthy gut microbiota is, however, evident as having microbial diversity, microbial gene richness and microbial functional activity [21]. Alterations in gut microbiota relate to antibiotic use, food preservatives, diet, and improvement in standards of living and hygiene [20]. Antibiotic use depletes the resident microbes, where exposure to antibiotic therapy in early life leads to long-term immune dysregulation and visceral hypersensitivity [40]. Dietary compounds, e.g., flavonoids, exert a major impact on the gut microbiome diversity typically in a positive manner, stimulating the growth of beneficial bacteria [21]. Importantly, the Western diet, which is low in fibre and high in a complex mixture of fats and simple sugars, is believed to reduce the microbial diversity or remove essential taxa from the human microbiota in Western populations [41]. Rodent-based studies demonstrated that a high-fat diet (60% fat) decreases the number of bacterial species and alters the diversity of the intestinal microbiota [42]. Such a diet is associated with obesity, low grade inflammation and metabolic disorders, such as type 2 diabetes (T2D). Indeed, more than 80% of patients with T2D in the Western world are obese with altered gut microbiota, inflammation, and gut barrier disruption [42]. The pro-inflammatory bacterial lipopolysaccharide (LPS) toxin has been identified in the systemic circulation of obese and type 2 diabetic patients and is believed to originate from the intestinal microbiota [42]. Importantly, the LPS toxin release from Gram negative bacteria impacts on gut-barrier function, adipose inflammation, intestinal glucose absorption, blood glucose, insulin and incretins, impacting on the prevalence of metabolic disorders [32]. Reduced levels of intestinal *Firmicutes* species has been reported in persons with T2D and obesity [23]. Studies demonstrate the reduced diversity evident in obese hosts where alterations in relative abundance of the major phyla *Firmicutes* and *Bacteroidetes* is present often with an overabundance of pathogenic microorganisms [43]. Studies have shown that in cases of severe intestinal inflammation, as observed in IBD and cancer, increased abundance of *Enterobacteriaceae* are present [32]. Low or non-calorie artificial sweeteners, e.g., saccharin, sucralose and aspartame also negatively impact microbial diversity [44] and have been associated with the induction of glucose intolerance in mice [21]. Non-antimicrobial therapeutics are also known to impact the resident microbiota. Indeed, studies have shown that commonly used drugs, including antipsychotics, proton-pump inhibitors (PPIs), hormones and anticancer drugs have a deleterious impact on microbial species of the GIT [21]. Opioid-induced dysbiosis has been associated with dysbiosis associated disease states and opioid tolerance [11]. Interestingly, Metformin, a therapeutic for the treatment of T2D, has recently been shown to alter the gut microbial diversity [21]. The impact of environmentally polluting chemicals, including heavy metals (mercury and lead), antimicrobial nanoparticles (silver) and endocrine disrupting chemicals (bisphenol A, phthalates) may also contribute to gut dysbiosis [24]. Studies have demonstrated that chronic exposure to low doses of the insecticide Chlorpyrifos altered the gut diversity in rodents and simulated human intestinal microbiota preparation [21].

### 2.2. How the Microbiota Interacts with the Nervous System and Induces Pain

The brain is known to regulate gastrointestinal function via the enteric nervous system and the vagal nerve (VN). The VN connects the internal organs (visceral organs) with the brain and is composed of sensory (afferent) and motor (efferent) neurons being part of the parasympathetic branch of the autonomic nervous system (ANS) where it regulates many critical bodily functions including mood, immune response, inflammation, digestion and heart rate [45]. This gut-brain axis (GBA) relies on bidirectional communication between the GIT enteric nervous system, hypothalamic pituitary adrenal (HPA) axis and the CNS [46]. The GBA has varied routes of communication utilising endocrine, immune and neural mechanisms, which the microbiota is capable of agonising to influence the CNS and can lead to pain associated with neuroinflammation [47]. The GBA is influenced by the resident microbiota via the secretion of neuroactive biologics, including serotonin, dopamine, acetylcholine and γ-aminobutyric acid (GABA) [5]. Specifically, intestinal species, including *Escherichia* spp. and *Lactobacillus* spp., are known to synthesize GABA [30] and acetylcholine [48]. Intestinal *E. coli* can produce norepinephrine, 5-hydroxytryptamine (5-HT) and dopamine, while *Streptococci* and *Enterococci* produce 5-HT [48]. Enterochromaffin cells allow for communication between the microbes and the CNS as they are agonised by microbial products and secrete 5-HT into the lamina propria and blood system [27]. Some microbial species present in the gut can also induce the release of intestinal peptides and hormones from host enteroendocrine cells and immune modulators, including cytokine and chemokines [47]. Central sensitization is associated with neuroinflammation via the activation of inflammatory immune cells and inflammatory molecules (cytokine, chemokines). Such immune mediators interact with nociceptors on the CNS and PNS, allowing for alterations in pain pathways resulting in neurological pain [5].

The afferent fibres of the vagal nerve present in the gastrointestinal wall are also stimulated by bacterial metabolites sugars, short chain fatty acids and GABA [5]. The HPA axis is a part of the limbic system of the brain associated with memory and emotional processing and is responsible for stress responses being activated by pro-inflammatory cytokines [46]. In response to stress triggers, the HPA axis stimulates the secretion of the corticotropin-releasing factor (CRF) from the hypothalamus, which in turn stimulates adrenocorticotropic hormone (ACTH) secretion from the pituitary gland, ultimately leading to the release of cortisol from the adrenal glands [49]. Pro-inflammatory cytokines can stimulate the HPA axis [50]. Microbiota produce the SCFAs propionate, butyrate and acetate in the GIT, which are absorbed by the intestinal epithelial cells or colonocytes and regulate cellular processes, including gene expression, chemotaxis, differentiation, proliferation and apoptosis [14]. SCFAs act as an energy source in the body, but can also act as agonists of G protein-coupled receptors, free fatty acid receptor 2 (FFAR2, GPR43) and FFAR3 (GPR41) inducing a signalling response, regulating metabolism and satiety [51]. Studies show that microbial SCFAs may impact neurotransmitters, including glutamate, glutamine and GABA, the synthesis of dopamine, noradrenaline and adrenaline, and also to regulate the expression of tryptophan 5-hydroxylase essential for serotonin synthesis. [27]. Tryptophan crosses the BBB and is available for serotonin synthesis in the brain [52]. Serotonin over-production is believed to lead to a dysregulation of the GBA involving the CNS and PNS, resulting in chronic pain conditions including IBS [5]. Additionally, SCFAs stimulate mucosal release of endocrine agents from the intestinal endocrine cells and inhibit histone deacetylase activity which has effects in the peripheral and central nervous systems [27]. Butyrate inhibits pro-inflammatory pathways and also plays a key role in preventing systemic exposure to intestinal antigens [52]. Importantly, studies in germ-free mice have shown that the diversity of the microbiota is essential for neurological processes, including development, myelination, neurogenesis and microglia activation [53]. Intestinal bacterial species may also produce neurotoxic molecules, e.g., d-lactate, ammonia and neurotoxins, e.g., *Clostridium tetani* producing botulinum, which can enter the CNS via systemic or extrinsic afferent nerve fibres, leading to neuronal damage [48]. Additionally, the LPS toxin, which is a constituent of the cell membrane of gram-negative bacteria, may induce neuro-inflammatory reactions if it gains entry to the systemic circulation from the GIT; however, LPS is unable to penetrate the BBB in healthy persons [23].

## 3. Chronic Pain Conditions

As chronic pain conditions increase globally, it is imperative to determine causative mechanisms of pain conditions. While somatic conditions of chronic pain are somewhat more readily treated, certain chronic pain conditions lacking an obvious cause are difficult to treat and manifest as life-long conditions having a drastic impact on the patient. Additionally, due to the co-morbid nature of pain conditions, the multifaceted nature of the immune system, nervous system and commensal microbiota, it is important to develop a better understanding of their interconnected relationship.

### 3.1. Neurological Conditions

The relationship between the gut microbiota on many chronic conditions, particularly neurodegenerative disease, is an area of interest as studies have shown that the microbiota can modulate the endocrine, nervous and immune systems [43]. The crosstalk between the gut and the brain helps maintain a healthy neurological state as the enteric bacteria impact on brain development and behaviour [30]. Neurological degenerative diseases include Parkinson disease (PD), Alzheimer disease (AD), multiple sclerosis (MS), Motor neuron disease, Amyotrophic lateral sclerosis (ALS) and Huntington’s disease, amongst others [53,54]. Neurodegenerative diseases are caused by a progressive loss of neurons in the CNS, with no cure and limited treatment options available. Interestingly the BBB displays increased permeability in neurodegenerative diseases [23,55]. An abundance of pro-inflammatory microbial species in the gut has been associated with the inflammation and neuroinflammation present in neurodegenerative diseases [56,57]. Drastic alterations in gut commensal bacteria have been seen in patients suffering from AD, PD, MS and additional neurological conditions of Autism, schizophrenia and major depression disorder (MDD) [43]. The intestinal microbiota also modulates a range of neurotrophins, e.g., brain-derived neurotrophic factor (BDNF) and proteins involved in brain development and plasticity [40]. GIT dysbiosis can induce local immune activation leading to systemic inflammation, neuroinflammation and changes in CNS functioning [34]. Activation of neurogenic inflammation via innate immune mechanisms initiates and maintains neuropathic pain as afferent nociceptive nerves communicate with the immune system [58]. Importantly, chronic pain, including radicular neuropathic pain and central neuropathic pain is present in many neurological diseases, affecting 20–40% of neurology patients [59].

#### 3.1.1. Parkinson’s Disease

Studies have shown that alterations in the composition of the microbiota is present in patients with PD [53]. The gastric organism *H. pylori* and the aetiological agent of gastric ulcers has been associated with PD since the 1960s, where the successful treatment of infection alleviates PD symptoms [30]. Alterations in intestinal microbiota in PD patients can lead to the accumulation of the protein α-synuclein and the elevated activation of microglia in brain neurons [43]. Genetic studies on stool samples from PD patients determined a lower diversity of SCFA-producing bacteria, including *Blauia*, *Coprococcus* and *Roseburia*, compared to healthy controls [52]. Furthermore, an increase in *Enterobacteriaceae* species was established to be directly proportional to the severity of symptoms associated with stability, gait and rigidity [60]. Importantly, when germ-free mice were colonised via faecal transplantation with samples from PD patients, they developed classic PD symptoms, including motor issues, behavioural issues and neuroinflammation, which improved with antibiotic therapy [53]. Studies have also demonstrated that some GIT species, including *Lactobacillus* and *Enterococcus faecalis*, can affect the metabolism of the therapeutic Levodopa through increased tyrosine decarboxylase gene expression [61]. The diversity of non-bacterial species, such as the eukaryotic yeast, is also impaired in PD patients, with important species lacking as identified by genetic analysis [48]. Inflammation in the GIT is recognised as a contributor to PD, where IBD patients are at higher risk of developing PD. Biomarkers of intestinal inflammation calprotectin and zonulin, a biomarker of intestinal permeability, are elevated in PD patients [48].

#### 3.1.2. Alzheimer’s Disease

While the exact aetiology of AD has yet to be established, it is known that AD is a manifestation of cell loss, increased activation of signalling pathways, amyloid-β (Aβ) deposits, mitochondrial dysfunction, chronic oxidative stress, impaired energy metabolism and DNA damage [57]. Studies have determined that AD patients possess gut microbiota with significant changes in the composition of the intestinal microbiome or dysbiosis compared to healthy persons, where reduced levels of *Actinobacteria* (*Bifidobacterium*) and *Firmicutes* phylum’s were evident [23]. Gut permeability and leaky gut induced by dysbiosis is speculated to impact on disease development in obese AD persons leading to the systemic inflammation and neuroinflammation seen in AD patients [23]. Importantly, the relationship between pathogenic *E. coli* and *Shigella* species inducing inflammation and a reduced presence of anti-inflammatory *E. rectale* has been associated with peripheral inflammation, cognitive issues and brain amyloidosis, which may induce neurodegeneration in AD patients [62]. AD patients also display lower diversity of butyrate-producing bacteria and increased presence of proinflammatory species [63]. Post-mortem studies on AD patients have shown that the hippocampus and cortex possess LPS toxin levels up to 3 times that of non-AD patients, indicating a causative link between BBB permeability to LPS toxin in AD patients [23]. Gut microbiota-derived amyloids produced by intestinal *Streptomyces*, *Bacillus*, *Pseudomonas*, *Klebsiella* and *Staphylococcus* species, which are similar to CNS amyloids, may contribute to the pathology of AD. Such bacterial amyloids can induce cytokine production, inflammation, phagocytosis and innate immune reactions impacting CNS homoeostasis and pathology in persons with an increased permeability of the GIT and BBB [57]. The neurotransmitters GABA and acetylcholine (Table 3) are involved in the pathology of AD due to their role in neural signalling and plasticity; dysbiosis with a reduced diversity of species producing such neurotransmitters may impact on AD progression [56].

#### 3.1.3. Amyotrophic Lateral Sclerosis

Amyotrophic Lateral Sclerosis is a neuromuscular disease due to the progressive death of motor neurons and muscle atrophy [64]. Compared to PD and AD, there is less research available on the impact of dysbiosis on ALS. This fatal neurodegenerative disease does, however, also display evidence of altered gut biodiversity, as studies show dysbiosis leading to an increased prevalence of proinflammatory organisms in ALS patients [65]. Animal studies have demonstrated the impact of dysbiosis on ALS, where a reduced content of butyrate-producing bacteria resulted in gut permeability [53]. Animal studies have demonstrated intestinal dysbiosis in rodents prior to ALS manifestation, where leaky gut was also present [64]. Furthermore, alleviating dysbiosis improved ALS progression with worsening dysbiosis associated with more severe clinical symptoms [64]. The *Firmicutes* to *Bacteroidetes* ratio in ALS patients is an area of much investigation, of which, studies, however, report contracting results as described by Boddy et al. (2021) [66].

#### 3.1.4. Multiple Sclerosis

Multiple sclerosis is an inflammatory disease of the central nervous system where host T cells are involved in the destruction of nerve tissue or demyelination and disease progression. MS is both an autoimmune disease and neurodegenerative condition, with ca. 75% of patients suffering chronic pain [59]. Studies have shown that a reduced quantity of butyrate-producing bacteria, particularly *Firmicutes*, are associated with the pathogenesis of MS [52]. Loss of integrity and increased permeability of the BBB is also a characteristic of MS and may be resultant from dysbiosis-induced intestinal permeability or leaky gut [53]. Studies assessing bacterial diversity in MS patients identified variations in species, including *Bacteroides* (butyrate-producing), *Prevotella* and *Sutterella* with bacterial species *Streptococcus thermophilus* and *Eggerthella lenta* significantly increased in MS patients [29].

#### 3.1.5. Mental Health Disorders

Psychiatric disorders including mood disorders MDD and generalised anxiety disorder (GAD) are highly prevalent globally as standalone illness and as co-morbidities of many disease states. Major depression is one of the most common mood disorders, impacting patient quality of life and causing social disability, resulting in an increased risk of morbidity, mortality and suicide [27]. Mental health disorders are common co-morbidities of diseases, including neurodegenerative, FSS and autoimmune diseases [5]. Such disorders can also intensify pain signalling, leading to increased levels of pain in affected patients. Unfortunately, such comorbid conditions can also significantly delay the diagnosis of pain conditions [4]. Excess *Bacteroidetes* and *Proteobacteria* and a decrease in *Firmicute* species has been identified in patients suffering from MDD [30], which is commonly associated with chronic pain co-morbidities, including FSS [5]. Gut microbiota associated with serotonin production may impact on mood disorders as they lead to a dysregulation of serotonin, which is associated major depressive disorder (MDD), anxiety disorders and schizophrenia [55]. Alterations in the intestinal microbiota resulting in altered SCFA composition and reduced quantities of tryptophan, the serotonin precursor amongst other neurotransmitters, including dopamine, can induce depression [36]. Butyrate has evident anti-depressant activity; however, levels of anti-inflammatory SSFAs are decreased in depression animal models compared to controls [33]. Studies also suggest that ketamine, as a therapeutic for depression, may have effects via restoration of *Bifidobacterium* levels [33]. MDD and GAD are characterised by hyperactivity or dysregulation of the HPA axis. Proinflammatory cytokines IL-6 and TNF-a and endotoxins (LPS) produced by bacteria can induce this dysregulation [50]. Importantly, evidence of bacterial translocation and systemic inflammation is present in schizophrenia patients where increased cytokine levels correspond to worsening symptoms due to neuroinflammation [40]. A correlation between antibiotic use and mental health issues including depression, anxiety and psychosis exists, where altered brain function is a psychiatric side effect of antibiotic use, where risk increases with repeated exposure [40].

### 3.2. Autoimmune Conditions

Autoimmune diseases are diseases manifesting when a patient’s immune system attacks self-cells and tissues leading to inflammation, tissue damage and chronic pain. Autoimmunity is therefore, a failure of self vs. non-self-recognition with the synthesis of autoantibodies (antibodies specific to self-tissues) or by the generation of autoreactive T lymphocytes [28]. Currently, there are approximately 100 listed autoimmune diseases, with the most common being Coeliac disease, Rheumatoid arthritis (RA), Inflammatory bowel disease (Crohn’s and colitis), Graves’ disease, Hashimoto thyroiditis, MS, Systemic lupus erythematosus (SLE) and type 1 diabetes (T1D) [67]. Studies assessing the role of the GIT microbiota on local and systemic (gut-distal autoimmunity) autoimmune disease have established that dysbiosis contributes to many of the disease states in this category [43]. The production of anti-inflammatory SCFAs by resident bacteria is one pathway where a shortage of SCFA production has been identified in MS [67]. Immuno-mediated encephalitis, which manifests as inflammation of the CNS, is an emerging group of syndromes which are non-MS autoimmune diseases of the brain and CNS associated with high mortality [68]. Encephalitis manifest due to an abnormal antibody response against cell-surface, intracellular synaptic, or intraneuronal antigens [69]. Studies have shown that the microbiota of encephalitis patients differed from that of healthy controls with alterations in SCFA production evident [68]. A decrease in butyrate- and propionate-producing species, including *Faecalibacerium prausnitzii* and *A. muciniphila*, is common in autoimmune diseases [35]. Systemic autoimmune diseases, e.g., RA, SLE and dermatological diseases (psoriasis) are characterized by an unusual adaptive immune response to autoantigens, specifically autoreactive T cells, which are a main driver of disease pathogenesis [70]. The microbiota influences the development of T lymphocytes, including Th-2/Th-1 lymphocytes, where the balance of effector T cells is altered in dysbiosis and results in T cell mediated autoimmune reactions [28]. Molecular mimicry which results from shared epitopes between GIT microbes and self-proteins may also lead to autoreactive T cell activation and autoimmunity [26]. As such, cross-reactivity of the immune system to microbial antigens may trigger autoimmune activity [71]. Molecular mimicry can also result in pain via direct IgG-induced injury of nociceptive fibres [58]. Additionally, the GIT microbiota is associated with the development of B lymphocytes and the synthesis of antibodies particularly immunoglobulin A (IgA) which is targeted against thymus-dependent and independent antigens [28]. Importantly, both autoreactive T cells and autoantibody-producing B cells have been associated with the pathogenesis of SLE [26]. In SLE, patient’s periodontitis is an issue with ca. 65% of patients affected, periodontitis is associated with dysbiosis of oral microbiota where pathogenic species of *Fusobacterium nucleatum* and *Actinomyces naeslundii*, amongst others, are present [28]. Additionally, the pathogenic *Enterococcus gallinarum* has been detected in the livers of patients with SLE and autoimmune hepatitis related to translocation from the GIT [71]. In RA patients, studies have shown a deficit of *Haemophilus* spp., which negatively correlates with the quantity of serum autoantibodies; furthermore, *Lactobacillus salivarius* was over-abundant and correlated with increased severity of symptoms [26]. The autoimmune disease of the pancreas T1D is associated with increased intestinal permeability and leaky gut with an excess presence of pro inflammatory species in the GIT microbiota [72].

### 3.3. Functional Somatic Syndrome

Functional Somatic Syndrome are a category of chronic pain conditions which do not have a somatic aetiology; currently, the three most common FSS include irritable bowel syndrome (IBS), fibromyalgia (FM) and chronic fatigue syndrome/myalgic encephalomyelitis (CFS/ME) [5]. IBS is a chronic and often severe functional gastrointestinal disorder (FGID) associated with abdominal pain, cramping, myalgia, diarrhoea or constipation, bloating and poor quality of life [73]. While the exact aetiology of IBS remains unclear, this FSS manifests with changed GIT motility, visceral hypersensitivity due to increased permeability [74], post infectious reactivity, altered GBA interaction, dysbiosis, bacterial overgrowth, food sensitivity, carbohydrate malabsorption and localised inflammation [73]. Additionally, similar symptoms of pain, etc., present in the varying forms of IBS, i.e., diarrhoea-predominant, constipation-predominant and mixed stool may be related to different microbiota and, as such, have a different aetiology [75]. Studies investigating the impact of dysbiosis on IBS have described an abundance of species including *Firmicutes*, *Streptococcus*, *Veillonella* and *Enterobacteriaceae*, or decreased *Bifidobacterium* [75]. Interestingly, studies have identified the presence of *E. coli* and *Ruminococcus gnavus* biofilms in the mucosa in IBS patients [76]. Other studies on IBS patient microbiota identified a decrease in microbial diversity, with severe symptoms in the presence of *Clostridiales* species and/or methanogenic microbes with an abundance of *Methanobravibacter smithii* present in constipation-predominant IBS [74]. The presence of methane-producing bacteria is associated with IBS constipation type with a decreased presence of SCFA-producing bacteria, including *Erysipelotrichaceae* and *Ruminococcaceae* [77]. Alterations in SCFA concentrations of butyrate and propionate is typically evident in IBS patients compared to controls [78]. Dysbiosis altering enteric serotonin production has been investigated in IBS where intestinal serotonin may induce visceral hypersensitivity, alter mucosal and gut permeability, activate the immune system, and induce inflammation manifesting as the symptoms of IBS [76]. Chronic low-grade inflammation and immune cell activation is associated with the IBS pathogenesis [79].

Fibromyalgia is a frequent co-morbidity of IBS and RA patients with overlapping symptoms present and is also associated with a high frequency (ca. 80%) of mental health comorbidities anxiety and depression [80]. While the pathophysiology of FM remains undetermined, FM manifests with generalized chronic musculoskeletal pain, cognitive issues, fatigue, asthenia, mood disorders, altered sleep patterns and FGIDs [81]. In FM patients, low-grade inflammation is often present via an increase in pro-inflammatory cytokines interleukin (IL)-6 and IL-8 [82]. This chronic low-grade inflammation sensitizes the neurons involved in detecting pain, transmitting pain signals and representing pain in the CNS, making them more excitable and sensitive [5]. Ultimately, FM is a condition of altered CNS nociceptive processing, altered peripheral nociception and systemic inflammation [83]. Dysbiosis and, particularly, small intestinal bacterial overgrowth (SIBO) due to the colonization of the distal small bowel with colonic bacteria, has been associated with FM [81]. Species including *F. prausnitzii*, *B. uniformis*, *P. copri* and *Blautia faecis*. *F. prausnitzii* are butyrate-producing microbes decreased in intestinal disease states and in FM patients as evident by the presence of circulating SCFA. *F. prausnitzii* is believed to have anti-nociceptive and anti-inflammatory effects and to improve barrier function [83]. Importantly, in FM patients, an abundance of butyrate producers (*I. butyriciproducens*, *F. plautii*, *B. desmolans*, *E. tayi*, *Parabacteroides merdae* and *E. massiliensis)* has been detected [84]. Several studies assessing the role of dysbiosis in FM pain describe leaky gut, LPS toxin, SCFAs and activation of the intestinal immune system promoting systemic inflammation in patients. Additional studies identified alterations in dysbiosis in FM patients [85].

CFS/ME is a chronic debilitating condition with symptoms of fatigue, post-exertional malaise, cognitive issues, muscular pain, GIT dysfunction, immune abnormalities and sleep disturbances [5]. Mitochondrial dysfunction and viral infections are believed to be key factors in the pathophysiology of CFS [86]. CFS patients display symptoms commonly observed in FM patients where an altered microbiota is also evident [83]. For example, *F. prausnitzii* has been shown to be depleted in CFS patients. Studies described have shown an abundance of *Enterobacteriaceae*, *Enterococcus* spp., *Streptococcus* spp. and a deficit of *Bifidobacteria* spp. and *Firmicutes* present [87]. The studies of Franklin et al. (2019) have suggested a link between reduced diversity and the lack of cardiac and respiratory fitness present in CFS patients [88]. Intestinal permeability and bacterial translocation to the systemic circulation is also believed to play a role in the pathogenesis of CFS, which may induce the increased levels of IgA, IgM and LPS present in patients [89]. The research of Lupo et al. (2021) identified a microbial diversity in CFS patients similar to that observed in Alzheimer’s patients and a *Firmicutes*/*Bacteroidetes* ratio similarly observed in autoimmune conditions, including Chron’s disease and SLE and T2D [90]. It is important to note, however, that the alterations in microbial diversity in CFS patients is not as well understood, where inconsistency in bacterial species present remains an issue [86].

## 4. Conclusions

The impact of the intestinal microbiota on disease is an ongoing area of research. The development of 16S rRNA genetic sequencing technologies has allowed for the identification of commensal organisms constituting the human microbiota. Research has demonstrated the impact of diet, exposure to antimicrobial agents and exposure to many other therapeutics on intestinal microbial diversity as having negative impacts on health. While the microbiota varies amongst individuals, a healthy microbial diversity consists of many anti-inflammatory species and reduced pro-inflammatory and pathogenic species. The interaction between the human microbiota, the nervous system and the immune system plays a role in body homeostasis and the aetiology of many disease states. The relationship between the resident microbes and conditions of chronic pain is becoming better understood. Butyrate-producing *Bacteroides* and anti-inflammatory *Firmicutes* in the intestines appear to be key players in many conditions of chronic pain. In recent years, it has become acknowledged that the intestinal microbiota has an important role in visceral pain, opioid tolerance, altered peripheral nociception and systemic inflammation. It is important that the oral microbiota is not overlooked, however, as it is associated with the production of inflammatory cytokines, production of exotoxins and endotoxins, which can enter the systemic circulation and are involved in neurocognition. Additionally, the resident microbiota may impact on therapeutics administered in the treatment of disease states as seen with levodopa in the treatment of Parkinson disease. Due to the co-morbid nature of chronic pain conditions with overlapping symptoms present, the establishment of an underlying causative agent would greatly improve treatment options for long suffering patients. Establishing the definitive impact of dysbiosis on disease states, nociception and neuroinflammation will provide insights into disease pathogenesis and may lead to improved diagnostic and therapeutic options. Furthermore, it may allow for the identification of risk factors, triggers, e.g., dietary, stress and environmental factors which promote flare-ups in chronic pain patients manifesting ith non-somatic conditions, including FM, IBS and CFS.

## Figures and Tables

**Table 1 biomedicines-11-00748-t001:** Types of chronic pain, aetiology and clinical considerations.

Type of Pain	Aetiology	Clinical Considerations
Neuropathic pain/ Radiculopathy	Caused by a lesion or disease of the somatosensory system, including peripheral fibres and central neurons [15], not always associated with a response to external stimulus, pain signalling pathways are damaged.	Ca. 30% of neuropathic pain is caused by diabetes, common in cancer survivors and neurodegenerative diseases. Lumbar and cervical painful radiculopathies often cause neuropathic pain [15]. Sleep disturbances, anxiety and depression are frequent and severe in patients with neuropathic pain [16].
	Blocking of conduction along sensory and motor axons results in loss of nerve function [17,18], compressed or inflamed nerve route leads to loss of function.	Causing pain, loss of sensation, and motor function depending on the severity, lumbosacral radiculopathy is very common [19], e.g., herniated disc with resultant nerve root compression or spondylosis [19].
Radicular pain	Pain radiates from an inflamed or compressed nerve root, not from stimulation of peripheral nerve endings, is not nociceptive pain [17].	Irritation of a sensitized nerve root, usually of a prolapsed intervertebral disc, pain is typically lancinating, shooting down the leg [18].
Nociceptive pain—2 types	Resultant from tissue damage by physical, chemical or traumatic events, injuries or infections [4], pain signalling pathways are intact.	
1. Somatic pain	Peripheral areas of the body	Triggered by an acute injury or chronic disease.
2. Visceral pain	Internal organs	Frequently associated with nausea, vomiting, nervousness.

**Table 2 biomedicines-11-00748-t002:** Human bacterial microbiota per region of the GIT and local environmental conditions.

GIT Location	Environmental Conditions	Phyla/Species Present
Oral cavity	Aerobic, pH 6.2–7.5	*Firmicutes*, *Fusobacteria*, *Proteobacteria*, *Actinobacteria*, *Bacteroidetes*, *Chlamydiae*, *Chloroflexi*, *Spirochaetes*, *Synergistetes*, *Saccharibacteria*, *Gracilibacteria* [37].
Stomach	Aerobic, pH 3	*Helicobacter pylori*, acid-resistant bacterial strains *Streptococcus*, *Neisseria* and *Lactobacillus*—may be transient bacteria, *Firmicutes* and *Proteobacteria* are the most abundant phyla in gastric mucosal samples [38], *Streptococcus* and *Prevotella* [39].
Small intestine (Duodenum, Jejunum, Ileum)	Aerobic, pH 5.7–7.5 Ileum also has a mucous layer similar to the colon	Firmicutes and Proteobacteria as major phyla [32]. Ileum also houses anaerobes *Bacteroidia*, *Ruminococcaceae* and *Lachnospiraceae* [32].
Colon	Anaerobic, pH 6.7–8.5, dense mucous layer present	*Firmicutes (predominantly Ruminococcaceae* and *Lachnospiraceae)*, *Bacteroidetes*, *Actinobacteria*, *Proteobacteria* and *Verrucomicrobia* [32].

**Table 3 biomedicines-11-00748-t003:** Biologically active compounds produced by or stimulated by resident gut microbiota, their nerve function and associated conditions of chronic pain.

Type of Biologic	Active Biological	Nerve Function	Associated Conditions
Neurotransmitter	Serotonin	Major neurotransmitter of the CNS and is associated with cognitive abilities, mood, sleep and appetite [5], regulates nociception, motor activity and mood [54]	MDD, anxiety disorders, schizophrenia [55]
	Dopamine	Regulates motor control, reward and cognitive function [5]	PD, MDD, anxiety, RA, FM, CFS/ME, IBS [55]
	Acetylcholine	Neural cell signalling, plasticity and communication [56]	AD [56]
	GABA	Principal inhibitory neurotransmitter in CNS [54]	MDD, anxiety, RA, FM, CFS/ME, IBS [55]
	Glutamate	Principal excitatory neurotransmitter in the CNS [54]	
Hormone	CRF	Regulates stress response [5]	MDD, anxiety disorders, IBS, FM, CFS [5]
Metabolites	Short chain fatty acids—acetate, propionate, and butyrate [21,27]	Regulate the expression of tryptophan 5-hydroxylase, stimulate mucosal release of endocrine agents, inhibit histone deacetylase activity, role in microbiota-gut-brain crosstalk [27]	Behavioural and neurologic pathologies, MDD, AD, PD, MS and autism [27,32] ALS [43,53]
Immune modulators	Interferons, cytokines, interleukins	Regulate neurodevelopment, neuroinflammation, and synaptic transmission	FM, RA, CFS, IBS, MDD, anxiety, and sleep disorders [5,50] neurodegenerative disease [57]

## Data Availability

Not applicable.

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
