# Peer review of "The Association between Dysbiosis and Neurological Conditions Often Manifesting with Chronic Pain"

_biomedicines, 2023, doi:10.3390/biomedicines11030748_

Round 1

Reviewer 1 Report

Very interesting.

The title is "The impact of dysbiosis on conditions of chronic pain."

The parts about Parkinson, ALS, AD are out of the scope of this review. Or the title and adstract should be changed, or parts that have nothing to do with pain should be kept out.

About neuropathic pain, there is just only 1 reference 33, (Research suggests that in-4 creased abundance of Parcubacteria and lower Verrucomicrobia results in neuropathic pain 5 and anhedonia (loss of pleasure) in test rats [33].)

And little evidence on FM. Should the article then not be better written as a hypothesis, that low grade inflammaton through disbiosis, might enhance, or activate pain?

The title states something different.

Minor issues;

Abstract: WHO: write it full (World Health Organisationdiseases.

"In particular, a decreased abundance of small chain fatty 18 acid e.g., butyrate producing bacteria including Faecalibacterium, Firmicutes and some Bacteroides spp 19 is frequently evident." What is fequently evident? Please make this clear in the abstract.

Text: If you do not use in the text abbreviations, do not write them such as TCA's, NMDA OIH

page 2 line 14: "with autoimmune conditions e.g., rheumatoid arthritis, osteoarthritis, neuropathic pain," Osteoartritis is not an autoimmune condition. If you only want to write that RA is an autoimmune condition, please put "autoimmune conditions e.g., rheumatoid arthritis" in the end of the sentence.

Table 1: Radiculopathy is neurpathic pain. Should be under this chapter. Radicular pain is nerve nerve pain, and should be before nociceptive pain.

Reviewer 2 Report

In this review, the author discusses an important and timely topic: the link between gut bacterial malfunction and chronic pain. Although the author thoroughly describes the link between altered gut bacterial populations and Neurodegenerative diseases what is missing or not adequately emphasized is the direct link to chronic pain.

It´s true that all of these neurological conditions can be associated with chronic pain to varying degrees, but this does not mean that chronic pain and altered gut bacterial populations are linked. In other words, it seems, from the review, that bacterial flora is mainly related to specific symptoms and that chronic pain has to be considered separately.

I would suggest emphasizing and describe more specifically the link between altered gut bacteria and chronic pain.

 Minor points:

Some sentences have to be rephrased since they are not clear:

Pag 5 lines 34-35

Pag 6 linea 4-7

Pag 6 lines 20-23

Pag 7 lines 19-21

Round 2

Reviewer 2 Report

I fill satisfied with the solution to my doubts find out by the authors. Up to me the paper can be accepted

Author Response

Thank you for your review.